# Influence of Volumetric Damage Parameters on Patch Antenna Sensor-Based Damage Detection of Metallic Structure

**DOI:** 10.3390/s19143232

**Published:** 2019-07-23

**Authors:** Zhiping Liu, Hanjin Yu, Kai Zhou, Runfa Li, Qian Guo

**Affiliations:** 1School of Logistic Engineering, Wuhan University of Technology, 1178 Heping Ave, Wuhan 430063, China; 2Engineer Research Center of Logistic Technology and Equipment, Ministry of Education, Wuhan 430063, China

**Keywords:** patch antenna, crack, corrosion pit, resonant frequency

## Abstract

Antenna sensors have been employed for crack monitoring of metallic materials. Existing studies have mainly focused on the mathematical relationship between the surface crack length of metallic material and the resonant frequency. The influence of the crack depth on the sensor output and the difference of whether the crack is depth-penetrated remains unexplored. Therefore, in this work, a numerical simulation method was used to investigate the current density distribution characteristics of the ground plane (metallic material) with different crack geometric parameters. The data reveals that, compared with the crack length, the crack depth has a greater influence on the resonant frequency. The relationship between the frequency and the crack geometric parameters was discussed by characterizing the current density and sensor output under different crack lengths and depths. Therefore, the feasibility of monitoring another common damage of metallic materials, i.e., corrosion pit, was explored. Furthermore, the influences of crack and corrosion pit geometric parameters on the output results were validated by experiments.

## 1. Introduction

While operating under an alternating load and corrosive environment, metallic structures are prone to structural damage, such as cracks and corrosion pits. This damage often leads to failure or even destruction of the structure [1,2]. Studies focused on structural health monitoring (SHM) that allows detailed characterization of metallic structure damage are essential for ensuring safe and reliable operation of mechanical equipment and to reduce related maintenance costs. The detected response is used to combine characteristics of the system, thereby allowing damage assessment and location within the SHM system. The data acquisition and processing system used requires many sensors working in tandem [3]. As an emerging sensor capable of passive wireless interrogation, the patch antenna sensor is used for structural damage monitoring.

Huang [4] designed a sensor that can wirelessly monitor ground plane cracks to verify the feasibility of patch antenna sensors for crack detection. The results confirmed that such sensors can detect and monitor fatigue crack growth with a sub-millimeter resolution [5,6,7,8]. Moreover, the linear dependence of the normalized resonant frequency and the frequency ratio $\left(f_{01}/f_{10}\right)$ on the crack tip location was revealed. Deshmukh [9] demonstrated the feasibility of using the sensor to monitor the crack length. Since then, the number of studies based on this topic has increased gradually. Through fatigue crack tests, various studies have evaluated the variation in the crack length and sensor resonant frequency during crack propagation. The results showed that the frequency decreases with an increasing crack length [9,10,11,12,13,14]. Based on the results obtained of the microstrip patch, Mohammad [15] discussed the effect of crack parameters on the resonant frequency of sensors from the perspective of surface current density distribution. Damage samples with different parameters were fabricated to investigate the influence of the crack length, location, and orientation on the frequency. Considering the influence of cracks on the current density distribution of the antenna radiation pattern, Ke [16] proposed an algorithm, which introduced a flux coefficient, for predicting the resonant frequency shift caused by random cracks. Simulations and experiments were performed to validate the accuracy of the algorithm.

In addition to the crack length, the crack width and depth may also have an influence on the resonant frequency. Cho [17] used patch antenna sensors to monitor the influence of the crack width on the resonant frequency during depth-penetrated type crack propagation. The results revealed that when the crack width increases, the resonant frequency shifts. For crack widths of 0 to 30 mils (0.76 mm), the frequency decreases monotonically. Yi [18] discussed the influence of the crack width, characterizing prefabricated depth-penetrated type cracks on the resonant frequency during crack propagation under fatigue test conditions. The results revealed that the frequency increases at small crack widths, but decreases when the width increases to a critical value. However, the patch is destroyed and the crack identification function is lost at certain levels of crack propagation during the fatigue crack test. The crack parameters measured in the late stage of fatigue testing are invalid (rather than actual) parameters. Therefore, the influence of geometric parameters on resonant frequency during crack propagation is difficult to explain. Zhang [19] investigated the influence of crack depth variation on the resonant frequency, based on observations of the surface current distribution associated with the patch and the electric field distribution in the vertical patch direction. The experimental results revealed that the frequency exhibits an approximately linear dependence on the depth. However, a length-penetrated type crack, which exceeded the detection range of the patch, was employed. Therefore, the dependence of the resonant frequency on the depth of length non-penetrating type cracks remains unexplained. In addition, the interaction between the various crack parameters and the variation in the frequency when multiple crack parameters change simultaneously is yet to be explored. 

Structures operating in a corrosive environment are prone to damage, such as corrosion pits. The influence of corrosion pits, which are one type volumetric structural damage, on the surface current density of the ground plane is similar to that of cracks. Previous studies have mainly focused on the piezoelectric ceramic transducer [20] and eddy current detection principle [21] for the detection of corrosion pits. However, research on the effective detection of corrosion pits is insufficient.

Based on the engineering requirements and research status, the aims of the present work were to: (1) Explore the influence of whether crack is depth-penetrated on the law that crack length affects resonant frequency; (2) discuss the influence of a volumetric corrosion pit’s geometric parameters on the resonant frequency; and (3) propose a hypothesis for qualitatively explaining the influence of volumetric damage’s geometric parameters on the resonant frequency.

This paper is organized as follows: The second part presents the theoretical analysis, design of the patch antenna sensor, and the proposition of the current path bypassing assumption. The third part presents the numerical simulation considering the influence of the volumetric damage geometric parameters on the current density distribution of ground plane, resonant frequency of the sensor, and the interaction between the geometric parameters describing the damage. In the fourth part, the tests for verifying the accuracy of the simulation results are presented. The fifth part provides a summary and topics for future work.

## 2. Theoretical Analysis

### 2.1. Mechanism of Crack Monitoring

The patch antenna sensor consists of a radiating patch, substrate, and ground plane (see Figure 1a). When the sensor is externally excited, a resonant cavity (see Figure 1b) is formed between the lower surface of the patch and the upper surface of the ground plane. The resonant frequency achieved by the antenna operating in the $TM_{mnp}$ mode is denoted as ${\left(f_{r}\right)}_{mnp}$. and can be calculated as follows:(1)$${\left(f_{r}\right)}_{mnp}=\frac{c}{2\pi \sqrt{\varepsilon_{re}}}\sqrt{{\left(\frac{m\pi }{h}\right)}^{2}+{\left(\frac{n\pi }{L}\right)}^{2}+{\left(\frac{p\pi }{W}\right)}^{2}},$$
where *c* is the speed of light in vacuum, $\varepsilon_{re}$ is the effective fringe of substrate, and *m*, *n*, and *p* represent the number of half cycles associated with the amount of electromagnetic field in three directions, respectively. 

The patch geometric parameters of the patch antenna sensor used in this work satisfies L>W>L/2>H. Therefore, the antenna operates under the $TM_{010}$ and $TM_{001}$ modes, corresponding to resonant frequencies $f_{010}$ and $f_{001}$ and current paths parallel to the long and the wide side of the patch, respectively. In these two resonant modes, the damage geometric parameters exert a similar influence on the resonant frequency [15] and, hence, for the convenience of analysis, the $TM_{010}$ mode was considered in this work. The relationship between the effective length of the patch and the resonant frequency is given as follows:(2)$$f_{010}=\frac{c}{2L_{eff}\sqrt{\varepsilon_{re}}},$$
where $L_{eff}$ is the effective length of the patch. For a given effective permittivity of the substrate and set of geometric parameters, the resonant frequency variation is only caused by the shift in the effective length of the patch.

To determine the influence of volumetric damage on the current path of the ground plane, the volumetric crack was taken as an example and the current path bypassing assumption is proposed as follows: The current path is completely distributed over the surface of the ground plane and the inner surface of the crack. Owing to the volumetric crack, the current path on the ground plane changes from one type to four types (see in Figure 2), namely: Current path 1: No crack interference and no change in current path length; current path 2: The current path is disturbed at the crack tip and bypasses the tip along the ground plane surface according to the shortest path principle; current path 3: The current path is disturbed on both sides of crack and passes from the crack length side to the crack width side, then to another crack length side, and finally returns to the ground plane surface. According to the shortest path principle, the current path should still bypass on the plane surface. However, the crack tip region can accommodate only a limited number of current paths; and current path 4: The current path is disturbed in the middle of the crack and passes through the crack length side to the crack bottom, then to another crack length side, and finally returns to the ground plane surface.

### 2.2. Design of Patch Antenna Sensor

The design of the microstrip patch antenna sensor mainly includes the design and parameter selection of the patch and the substrate. The length and width of the patch as well as the fringe and thickness of the substrate have a direct effect on the inherent resonant frequency of the sensor. The width, *W*, and length, *L*, of the patch are calculated as follows [8]:(3)$$\{\begin{array}{c} W=\frac{c}{2f\sqrt{\varepsilon_{re}}}+2\Delta W\\ L=\frac{c}{2f\sqrt{\varepsilon_{re}}}+2\Delta L \end{array},$$
where ΔW and ΔL are the equivalent radiation gap width and equivalent radiation gap length that must be considered for the fringe effect, respectively.

## 3. Simulation Analysis

### 3.1. Size of the Patch Antenna Sensor

The geometric parameters of the patch antenna sensor were designed based on Equation (3) listed in Section 2.2. The overall geometric parameters of the sensor and the related dimensional data are shown in Table 1 and Figure 3.

### 3.2. Simulation Results Analysis

The patch antenna sensor model shown in Figure 4 was built using the COMSOL Multiphysics simulation software. The default solver was set to: The number of iterations was 25, the tolerance factor was 1, and the residual factor was 1000. The distribution of the current density and the current path of the volumetric damage surface were observed, and the dependence of the simulation time on the number of grids was determined. To enable this observation and determination, a thin-walled entity (thickness: 0.05 mm) with the same shape as the damage was built around the damage to achieve a refined network division of damaged parts. The minimum grid element size of the thin-walled part of the crack was 0.05 mm, and the minimum grid element size of the patch and feeder part was 0.8 mm. The rest was subjected to free tetrahedral mesh division, the grid unit of which required a maximum unit size of 20 mm and the minimum unit size was 0.8 mm. By determining the geometric parameters of the cracks and corrosion pits, the resonant frequency values associated with different volumetric damage geometric parameters were calculated.

#### 3.2.1. Relationship between Volumetric Crack Geometric Parameters and Resonant Frequency

The vertical current path direction parameters have the most significant cutting effect on the current path (see Figure 2) and, hence only the crack length and depth were considered in the present study. The crack width was set to 0.5 mm. Furthermore, the relationship between the resonant frequency and the crack length associated with different crack depths (see Figure 5) can be obtained from the recorded simulation data. The variation in the sensitivity (frequency shift caused by the crack of per unit length; see Table 2) can also be obtained. The calculation formula of the sensitivity is shown as follows: The sensitivity value of the latter point is the difference of the resonant frequency between the two adjacent points divided by the difference of their geometric parameters (which could be the crack length and depth, corrosion pit radius, and depth), depicted as follows: (4)$${\mathrm{sensitivity}}_{x+1}=\frac{{\mathrm{frequency}}_{x+1}-{\mathrm{frequency}}_{x}}{{\mathrm{geometric}\;\mathrm{parameter}}_{x+1}-{\mathrm{geometric}\;\mathrm{parameer}}_{x}}.$$

As shown in Figure 5, the influence of the crack length on the resonant frequency gradually becomes more noticeable with increasing crack depth. When the depth increases from 5 to 6 mm (i.e., the crack transforms from non-penetrating to depth-penetrated), this influence gets drastic suddenly. When no crack penetrates through the ground plane (i.e., when the crack depth is 1 to 5 mm), the sensitivity fluctuation under different crack lengths is small (see Table 2). For a crack length of 24 mm, when the crack depth increases from 5 to 6 mm, the sensitivity increases significantly from 6.003 to 22.011 MHz/mm. The amplification of the crack depth leads, in general, to an improved sensitivity of the crack length identification, which intensifies sharply as the depth increases to the depth through the ground plane. Based on the current path bypassing assumption, with an increase in the crack depth, the current path length increment, the resonant frequency reduction, and the length identification sensitivity gradually magnify. When the depth rises to the depth through the ground plane, the current path bypassing the crack bottom varies to the crack side. This leads to a substantial amplification in the current path length and the identification sensitivity.

As shown in Figure 6, the influence of the crack depth on the resonant frequency becomes more prominent with the increasing crack length. However, when the depth rises from 5.5 to 6 mm, the crack penetrates the ground plane and the resonant frequency drops sharply. The depth identification sensitivity (associated with different crack lengths) decreases gradually at crack depths lower than the penetration depth of the ground plane (see Table 3). However, the depth identification sensitivity amplifies drastically when the crack penetrates the ground plane. As a crack length of 24 mm and without crack penetration of the plane (i.e., the crack depth is 1 to 5 mm), the sensitivity reduces from 30.016 to 16.008 MHz/mm. When the crack penetrates the plane, however, the sensitivity intensifies sharply to 188.096 MHz/mm. Overall, the amplification of the crack length leads to a magnification in the sensitivity of crack depth identification, with the sensitivity increasing sharply when the depth amplifies to the depth of ground plane. Based on the current path bypassing assumption, during the amplification of the crack length, the current path length increment, the resonant frequency reduction, and the length identification sensitivity increases gradually. The bypassing type of the partial current path varies, when the crack penetrates the ground plane, thereby resulting in a sudden fluctuation in the depth identification sensitivity.

#### 3.2.2. Influence of Volumetric Crack on Resonant Cavity EM Parameters

Taking the crack depth of 2 mm as an example, the middle layer of the ground plane (thickness: 6 mm) was selected as the XOY plane of Z = 0. The crack varies downward from Z = 3 mm, and fine meshing of the crack damage was obtained. A two-dimensional schematic of the current density distribution on each surface comprising the inner surface and the corresponding three-dimensional diagram are shown in Figure 7 and Figure 8.

As shown in Figure 8a, the overall current density shifts slightly in the crack length side. However, the current density near the upper and lower apex is abrupt and reversed, owing to the dense current distribution and the opposite current direction. The default meshing settings of the simulation software are uneven, thereby resulting in increased noise. As shown in Figure 8b, the current density varies only modestly in the width direction of the crack wide side. In the depth direction, the current density first amplifies to the maximum, and then decreases gradually to zero, as the depth increases. The current density in the crack bottom is (in general) small on both sides and large in the middle (see Figure 8c). 

The current density is more symmetric on the inner sides of the crack than in other regions. Therefore, the current density distributions along the four lines (a–d) in Figure 7 were compared with different crack depths (see Figure 9). 

As the crack depth increases, the length increment of the current paths bypassing the bottom gradually surpasses the paths around the side. The partial bypassing current paths vary from the bottom region to the side. As shown in Figure 9a, on the crack length side, the current density gradually decreases with the increasing crack depth. The skin depth, which results from the skin effect, indicates that the current distribution still exists within a certain depth of the ground plane. As shown in Figure 9b, on the crack width side, the current density intensifies sharply from zero to a maximum value, and then gradually drops to zero. Moreover, the rate of the current density reduction gradually decreases with the amplifying crack depth.

The current path on the crack bottom is only distributed from one length side to the other length side, as shown in Figure 10. Therefore, in the crack length direction, the value of the current density is zero on the crack bottom and is maximum in the middle region, which is characterized by a bowl shape. In the crack width direction, the current density fluctuation on the crack bottom is small.

#### 3.2.3. Influence of Corrosion Pit Geometric Parameters on Sensor Monitoring Performance

The recorded simulation data show the relationship between the resonant frequency and the corrosion pit radius (for different corrosion pit depths), as well as the sensitivity variation (see Figure 11 and Table 4).

As shown in Figure 11, the influence of the corrosion pit radius on the resonant frequency gradually becomes more significant with an increasing corrosion pit depth, and no abrupt changes occurred when the pit penetrated the ground plane. A small corrosion pit radius (i.e., 1 mm) is associated with a low sensitivity of radius identification (4.999–6.49995 MHz/mm, see Figure 4). The sensitivity of 19.0095 to 36.018 MHz/mm was realized for pit depths of 1 to 6 mm and a radius of 10 mm. Overall, the sensitivity gradually magnifies with an increasing corrosion pit radius and depth.

As shown in Figure 12, the influence of the corrosion pit depth on the resonant frequency gradually becomes more noticeable with an intensifying corrosion pit radius, but this influence decreases with an amplifying corrosion pit depth. However, when the depth varies from 5.5 to 6 mm, i.e., when the corrosion pit penetrates the ground plane, the frequency drops sharply. As shown in Table 5, the depth identification sensitivity associated with different corrosion pit radii gradually decreases for pit depths differing from the ground plane depth. Nevertheless, the sensitivity sharply intensifies when the pit penetrates the plane. When no penetration occurs at a pit radius of 10 mm (pit depth: 1–5 mm), the sensitivity reduces from 186.61 to 10.8 MHz/mm. However, when the pit penetrates the ground plane, the sensitivity steeply amplifies to 40.4 MHz/mm. Overall, amplification of the corrosion pit radius results in an enhanced corrosion depth sensitivity. The depth identification sensitivity amplifies sharply as the depth magnifies to the ground plane depth.

The surface current density distribution of the ground plane with a volumetric corrosion pit was revealed by fine meshing around the pit, as shown in Figure 13. The corrosion pit interior is a curved surface, and hence, visualization of the planar current density distribution was impossible. Therefore, the current density distributions along line a and b in Figure 13 are compared at different depths.

As the corrosion pit depth increases, the length increment of current paths bypassing the bottom gradually surpasses the paths around the side. Partial bypassing current paths shift from the bottom region to the side. As shown in Figure 14a, the current density along line a gradually decreases with increasing depth. The skin depth, which results from the skin effect, indicates that the current distribution still exists within a certain depth of the ground plane. As shown in Figure 14b, the current density along line b abruptly amplifies from zero to a maximum value, and then gradually decreases to zero. Furthermore, the rate of the current density reduction gradually drops with the magnifying depth of the corrosion pit.

## 4. Experiment Research

### 4.1. Preliminary Preparation

To validate the accuracy of the conclusion based on the relationship between the simulated resonant frequency and the volumetric damage geometric parameters, the volumetric damage identification experiment based on a patch antenna sensor was performed. Based on the simulation data and results, Q235 was selected as the ground plane material. The crack and corrosion pit damages were obtained via electric spark and drilling, respectively, based on the damage center coordinates. The patch antenna was processed by means of circuit board etching [16], and was pasted to the corresponding location of the ground plane via AB glue bonding. The vector network analyzer (VNA) model used in this work, i.e., Agilent E5061B, is valid for frequencies ranging from 100 kHz to 3 GHz. The corresponding study was performed on selected sample groups (see Table 6).

The sample geometric parameters data contained in the sample groups are shown in Table 7 and Table 8.

The required sample and experiment platform for the experimental study are shown in Figure 15.

### 4.2. Experimental Results Analysis

To verify the correctness of the simulation data, the resonant frequency data recorded during experiments were compared with the simulation data. The simulation and experiment data obtained for identification of the volumetric damage geometric parameters are compared in Table 9 and Table 10.

As shown in Table 9, the experimentally determined resonant frequency obtained from volumetric crack identification and the simulation data differs by 0.02% to 1.33%. The calculation formula of the error of the experimental data is shown as follows:(5)$$\mathrm{error}=\frac{\left|\mathrm{experimental}\;\mathrm{data}-\mathrm{simulation}\;\mathrm{data}\right|}{\mathrm{simulation}\;\mathrm{data}}\;\times \;100\%.$$

The correlation coefficients corresponding to different depths are $r_{1mm}=0.8608,r_{2mm}=0.9134,r_{3mm}=0.9781,r_{4mm}=0.9931,r_{5mm}=0.9887,r_{6mm}=0.9951$. The correlation coefficient is large for crack depths of ≥2 mm and hence, the experiment data are close to the simulation data. However, the correlation coefficient is small when the crack depth is 1 mm (see Figure 14 for a graphic comparison of the correlation data).

As shown in Figure 16, after removing the third and sixth abnormal data points, the rest of the experimental data approach the simulation data. Conclusions are drawn, based on the experimental data, and the results of cracks generated by the simulation data can be validated.

As shown in Table 10, the experimentally determined resonant frequency obtained from the volumetric corrosion pit and the simulation data differ by 0.11% to 3.08%. The correlation coefficients corresponding to different depths are $r_{1mm}=0.9968,r_{2mm}=0.9996,r_{3mm}=0.9979,r_{4mm}=0.9981,r_{5mm}=0.9986,r_{6mm}=0.9992$. The correlation coefficients vary with the depth and the experimental data are close to the simulation data. The experimental results can be verified and the relevant conclusions based on simulation of the corrosion pits can be validated.

## 5. Conclusions and Discussions

The length identification sensitivity gradually magnifies with an amplifying crack depth. However, when the crack penetrates the ground plane, the sensitivity decreases drastically. The depth sensitivity intensifies gradually with an increasing crack length, but decreases with an amplifying crack depth. However, when the crack penetrates the ground plane, the sensitivity drops steeply.The radius identification sensitivity magnifies gradually with an amplifying corrosion pit depth/radius. Similarly, the depth identification sensitivity gradually intensifies with an increasing pit radius, but decreases gradually with an increasing pit depth. However, when the corrosion pit penetrates the ground plane, the depth identification sensitivity drops sharply.Hypothesis proposal: Volumetric damage can lead to variations in the current path of the ground plane. This is mainly reflected in the fact that, owing to the damage, the current paths distributed on the plane surface vary along the ground plane surface, damage side, and damage bottom. Furthermore, this results in an amplification of the average current path length associated with the ground plane, ultimately leading to a reduction of the resonant frequency of the sensor.

## Figures and Tables

**Figure 1 sensors-19-03232-f001:**
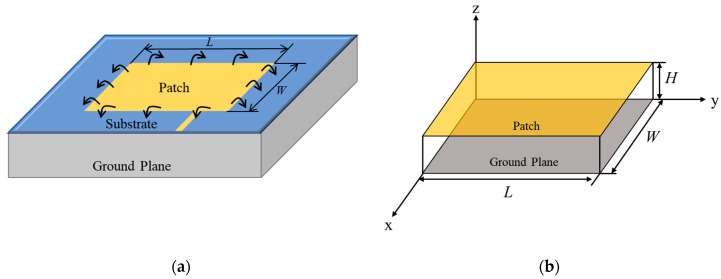
Crack monitoring mechanism of the patch antenna sensor. (**a**) Patch antenna senor structure and (**b**) resonant cavity model.

**Figure 2 sensors-19-03232-f002:**
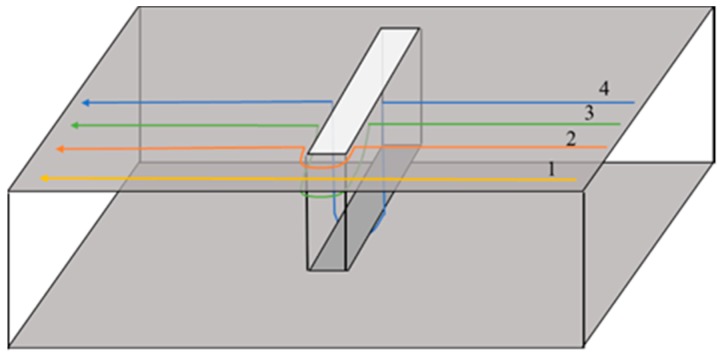
Current path classification on the ground plane.

**Figure 3 sensors-19-03232-f003:**
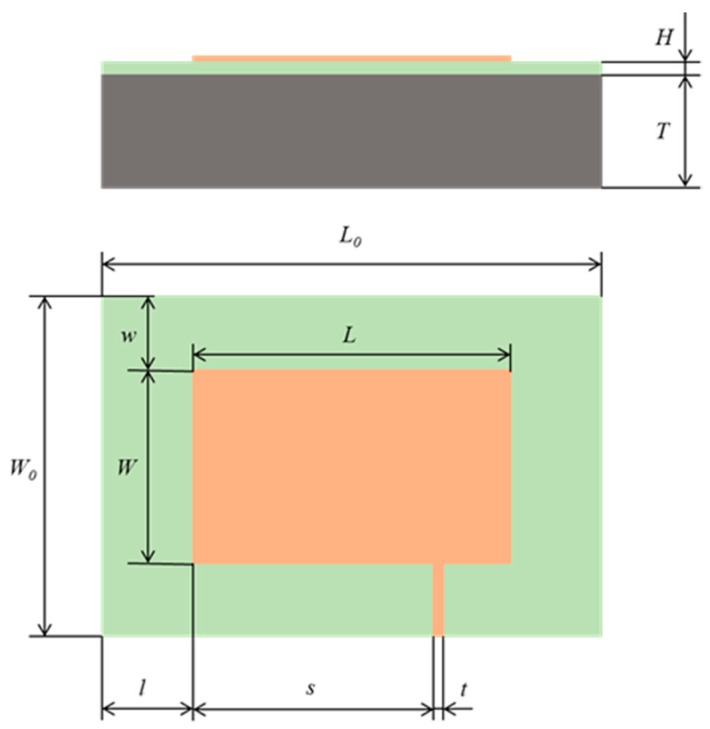
Dimension diagram of the patch antenna sensor.

**Figure 4 sensors-19-03232-f004:**
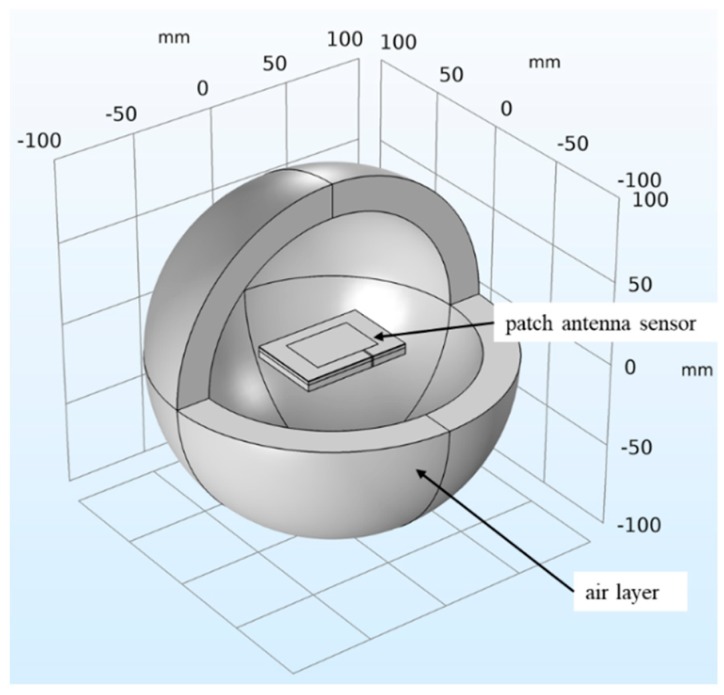
Simulation model of the patch antenna sensor.

**Figure 5 sensors-19-03232-f005:**
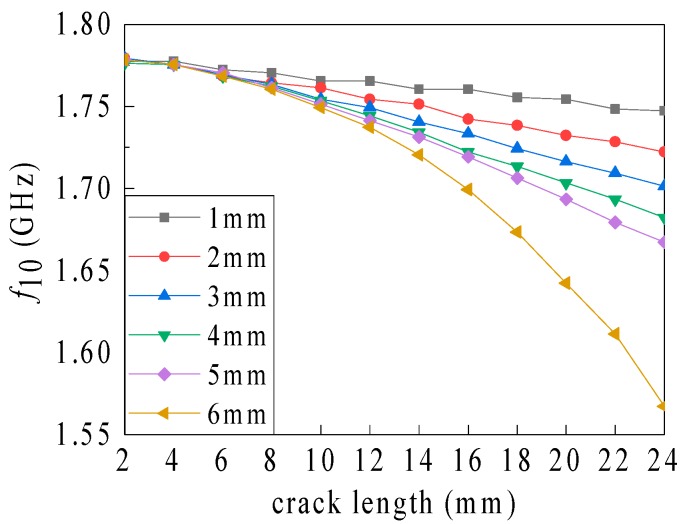
Dependence of the resonant frequency on the crack length associated with different crack depths.

**Figure 6 sensors-19-03232-f006:**
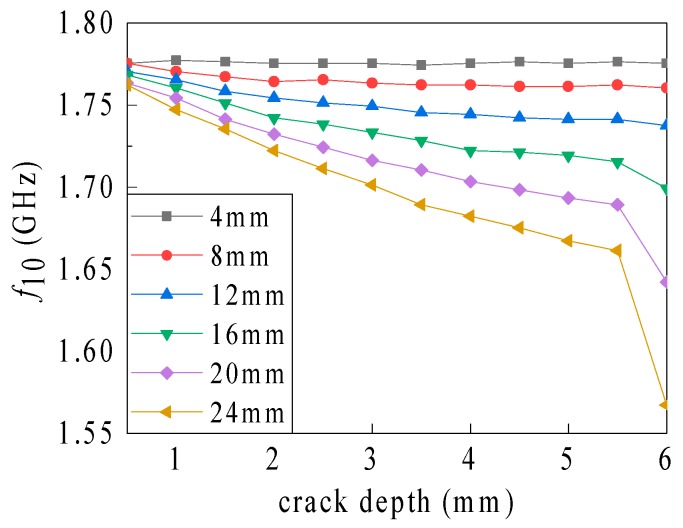
Relationship between the resonant frequency and the crack depth associated with different crack lengths.

**Figure 7 sensors-19-03232-f007:**
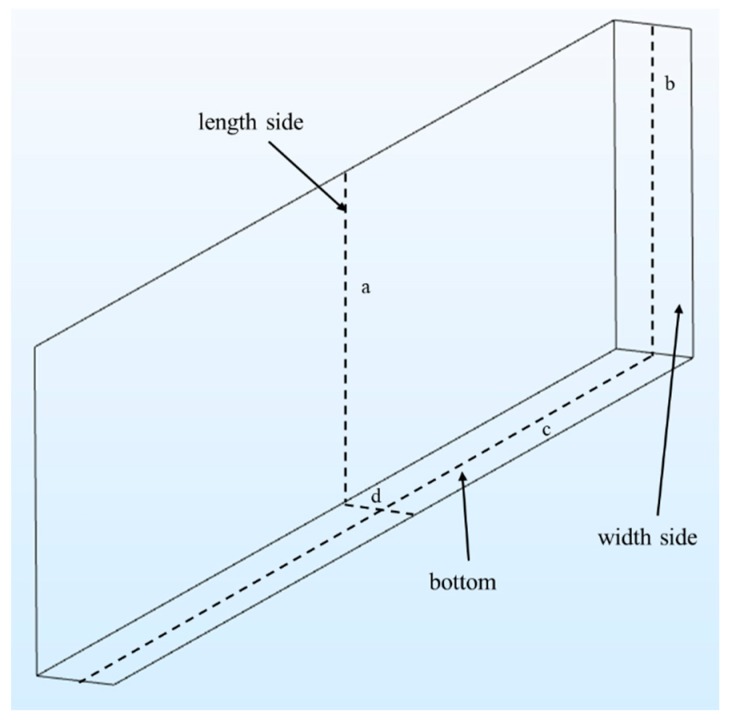
Current density distribution of the volumetric crack inner surface.

**Figure 8 sensors-19-03232-f008:**
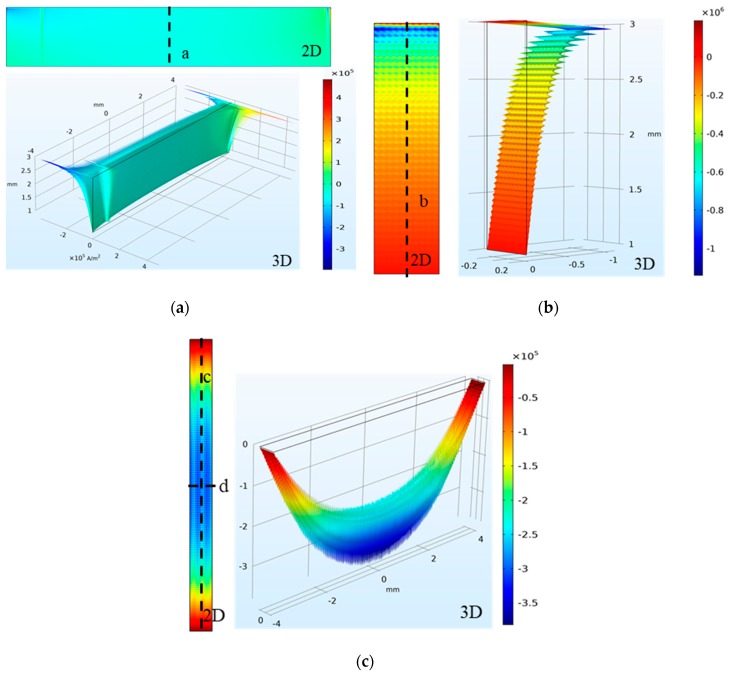
Two-dimensional (2D) and three-dimensional (3D) schematic showing the current density distribution on the (**a**) crack length side, (**b**) crack width side, and (**c**) crack bottom.

**Figure 9 sensors-19-03232-f009:**
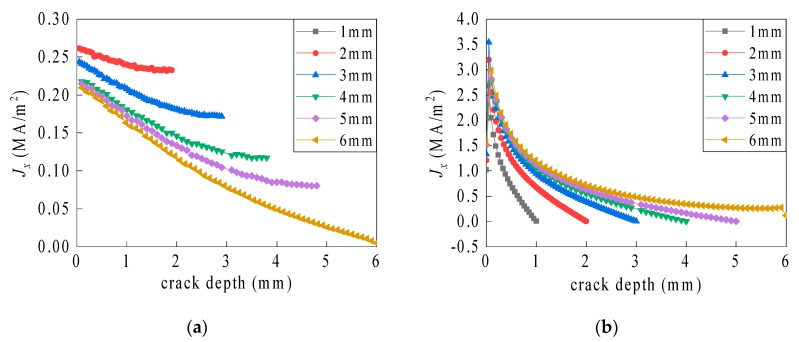
Current density distribution along a vertical axis of symmetry associated with the inner sides of the crack. (**a**) Line a and (**b**) line b.

**Figure 10 sensors-19-03232-f010:**
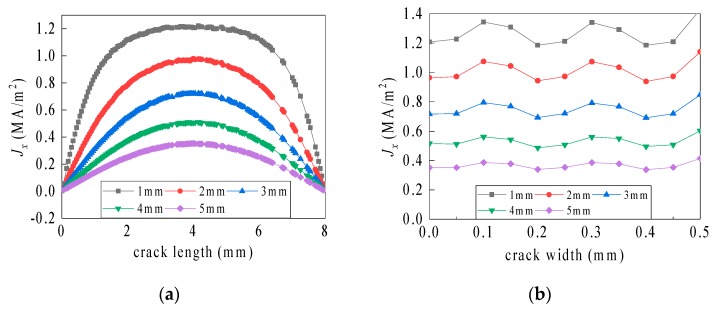
Current density distribution along the axis of symmetry on the crack bottom. (**a**) Line c and (**b**) line d.

**Figure 11 sensors-19-03232-f011:**
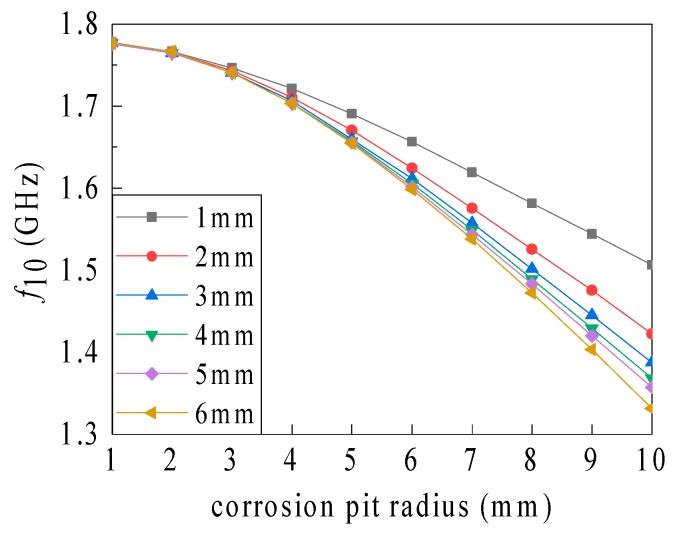
Influence of the volumetric corrosion pit radius on the resonant frequency.

**Figure 12 sensors-19-03232-f012:**
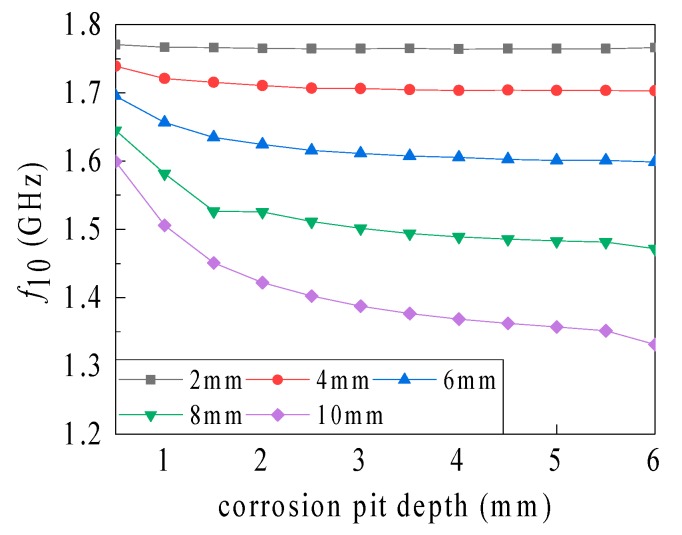
Influence of the volumetric corrosion pit depth on resonant frequency.

**Figure 13 sensors-19-03232-f013:**
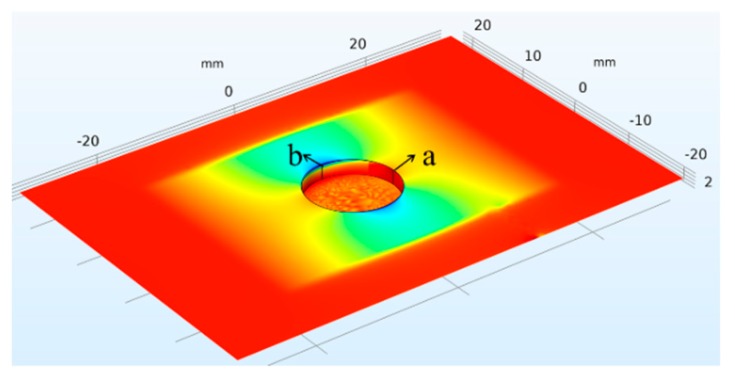
Current density distribution corresponding to the inner surface of the volumetric corrosion pit.

**Figure 14 sensors-19-03232-f014:**
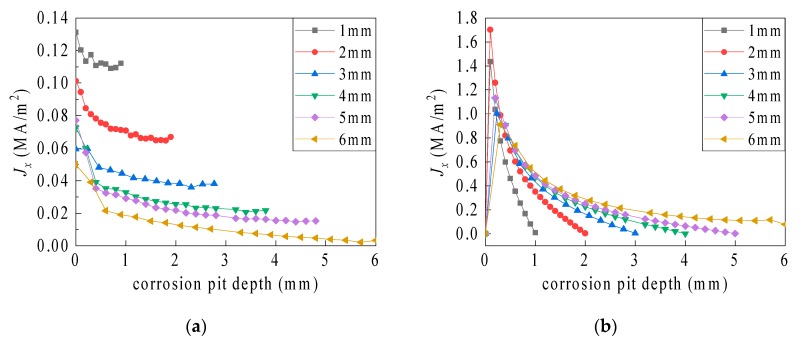
Current density distribution of the corrosion pit inner side. (**a**) Line a and (**b**) line b.

**Figure 15 sensors-19-03232-f015:**
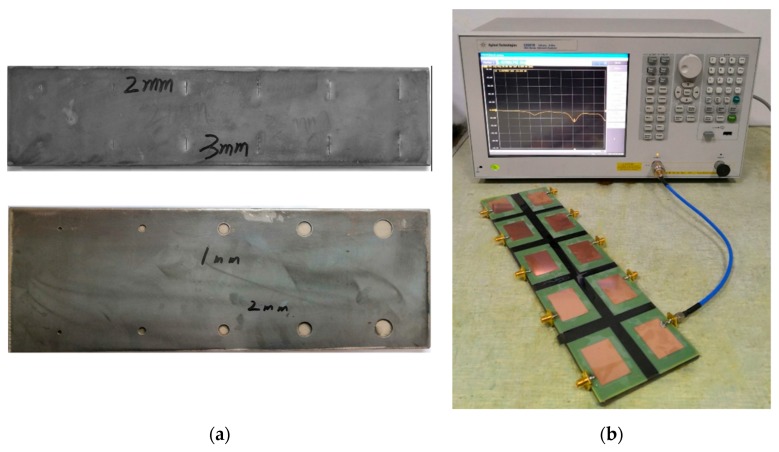
Experiment setup. (**a**) Sample and (**b**) experiment platform.

**Figure 16 sensors-19-03232-f016:**
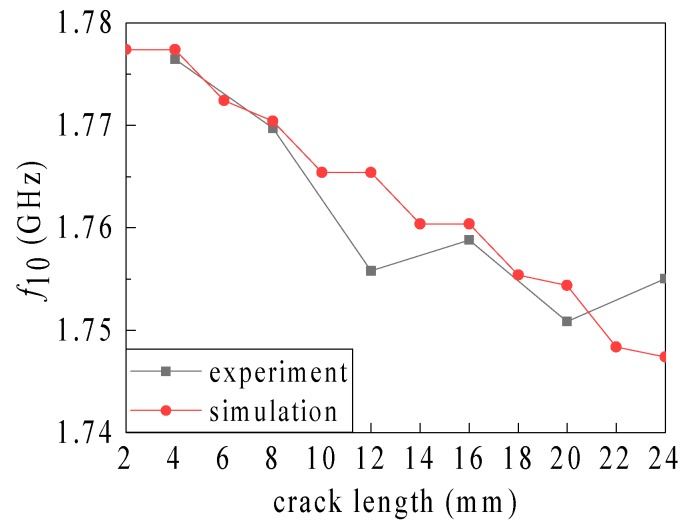
Comparison of simulation and experiment data corresponding to a crack depth of 1 mm.

**Table 1 sensors-19-03232-t001:** Geometric parameters of the patch antenna sensor.

Symbol	Parameter	Value
*H*	substrate thickness	0.5 mm
*T*	ground plane thickness	6 mm
*L* _0_	substrate length	64 mm
*L*	patch length	40 mm
*l*	horizontal distance of edges between patch and substrate	12 mm
*s*	horizontal distance of edges between feed and patch	29.5 mm
*t*	feed width	1 mm
*W* _0_	substrate width	44 mm
*W*	patch width	28 mm
*w*	vertical distance of edges between patch and substrate	8 mm

**Table 2 sensors-19-03232-t002:** Identification sensitivity of crack length.

Crack Depth (mm)	Sensitivity of Crack Length (MHz/mm)					
	4	8	12	16	20	24
1	0	−1.0005	0	0	−0.5005	−0.5000
2	−1.9970	−2.0010	−3.5045	−4.5025	−3.0015	−3.0015
3	−1.9970	−3.0015	−2.5010	−3.4995	−4.0020	−4.0025
4	−0.5000	−3.0015	−4.5025	−6.0030	−5.0025	−5.5030
5	−1.4965	−4.5020	−5.0025	−6.0030	−6.5030	−6.0030
6	−1.4965	−4.0020	−6.0030	−10.505	−15.5080	−22.0110

**Table 3 sensors-19-03232-t003:** Identification sensitivity of the crack depth.

Crack Length (mm)	Crack Depth Sensitivity (MHz/mm)					
	1	2	3	4	5	6
4	3.9820	−2.0000	0	2.0000	−1.9800	−1.9820
8	−10.0100	−6.0040	−4.0020	0	0	−4.0020
12	−10.0040	−8.0040	−4.0000	−2.0020	−2.0080	−7.9560
16	−15.9940	−18.0100	−9.9980	−12.0040	−4.0020	−32.0160
20	−18.0100	−18.0100	−16.0080	−14.0060	−10.0040	−94.0480
24	−30.0160	−26.0140	−20.0100	−14.0060	−16.0080	−188.0960

**Table 4 sensors-19-03232-t004:** Identification sensitivity of the corrosion pit radius.

Corrosion Pit Depth (mm)	Sensitivity of Corrosion Pit Radius (MHz/mm)				
	2	4	6	8	10
1	−5.4990	−12.5065	−17.0085	−19.0135	−19.0095
2	−5.50250	−16.5085	−23.0115	−25.0090	−26.5145
3	−5.9990	−17.0085	−24.0090	−28.0140	−29.0185
4	−6.4995	−18.5000	−26.0105	−30.0155	−30.5155
5	−5.5030	−18.5070	−27.0135	−30.0150	−31.5155
6	−4.9990	−18.5070	−28.0140	−32.6715	−36.0180

**Table 5 sensors-19-03232-t005:** Identification sensitivity of the corrosion pit depth.

Corrosion Pit Radius (mm)	Sensitivity of Corrosion Pit Depth (MHz/mm)					
	1	2	3	4	5	6
2	−7.8000	−2.2000	0	−1.8000	0	3.4000
4	−35.2020	−10.2020	−1.6000	−2.2000	−1.6000	−1.2000
6	−77.2040	−20.8000	−8.9920	−4.4000	−3.2000	−5.0000
8	−126.8060	−2.0020	−20.4000	−9.2020	−6.0000	−19.4100
10	−186.6100	−57.0020	−29.2020	−16.4020	−10.8000	−40.4000

**Table 6 sensors-19-03232-t006:** Sample group classification.

Research Purpose	Sample Groups
influence of crack length	G1, G2, G3, G4, G5, G6
influence of crack depth	G7, G8, G9, G10, G11, G12
influence of corrosion pit radius	G13, G14, G15, G16, G17, G18
influence of corrosion pit depth	G19, G20, G21, G22, G23

**Table 7 sensors-19-03232-t007:** Parameters associated with the crack specimen. (cl × cd, cw = 0.5 mm) (mm).

	G7	G8	G9	G10	G11	G12
G1	4 × 1	8 × 1	12 × 1	16 × 1	20 × 1	24 × 1
G2	4 × 2	8 × 2	12 × 2	16 × 2	20 × 2	24 × 2
G3	4 × 3	8 × 3	12 × 3	16 × 3	20 × 3	24 × 3
G4	4 × 4	8 × 4	12 × 4	16 × 4	20 × 4	24 × 4
G5	4 × 5	8 × 5	12 × 5	16 × 5	20 × 5	24 × 5
G6	4 × 6	8 × 6	12 × 6	16 × 6	20 × 6	24 × 6

**Table 8 sensors-19-03232-t008:** Parameters associated with the corrosion pit specimen. (ϕpr × pd) (mm).

	G19	G20	G21	G22	G23
G13	ϕ2 × 1	ϕ4 × 1	ϕ6 × 1	ϕ8 × 1	ϕ10 × 1
G14	ϕ2 × 2	ϕ4 × 2	ϕ6 × 2	ϕ8 × 2	ϕ10 × 2
G15	ϕ2 × 3	ϕ4 × 3	ϕ6 × 3	ϕ8 × 3	ϕ10 × 3
G16	ϕ2 × 4	ϕ4 × 4	ϕ6 × 4	ϕ8 × 4	ϕ10 × 4
G17	ϕ2 × 5	ϕ4 × 5	ϕ6 × 5	ϕ8 × 5	ϕ10 × 5
G18	ϕ2 × 6	ϕ4 × 6	ϕ6 × 6	ϕ8 × 6	ϕ10 × 6

**Table 9 sensors-19-03232-t009:** Comparison of simulation and experiment data associated with the volumetric crack.

	Length (mm)	4	8	12	16	20	24
Depth (mm)							
1	simulation	1.7774	1.7704	1.7654	1.7604	1.7544	1.7474
	experiment	1.7764	1.7697	1.7558	1.7588	1.7508	1.7550
	error	0.05%	0.04%	0.54%	0.09%	0.20%	0.44%
2	simulation	1.7754	1.7644	1.7544	1.7424	1.7324	1.7224
	experiment	1.7706	1.7629	1.7569	1.7307	1.7449	1.7248
	error	0.27%	0.09%	0.14%	0.67%	0.72%	0.14%
3	simulation	1.7754	1.7634	1.7494	1.7334	1.7164	1.7014
	experiment	1.7708	1.7627	1.7469	1.7398	1.7088	1.7088
	error	0.26%	0.04%	0.14%	0.37%	0.44%	0.44%
4	simulation	1.7754	1.7624	1.7444	1.7224	1.7034	1.6823
	experiment	1.7725	1.7627	1.7406	1.7207	1.6969	1.6610
	error	0.16%	0.02%	0.21%	0.09%	0.38%	1.27%
5	simulation	1.7754	1.7614	1.7414	1.7194	1.6934	1.6673
	experiment	1.7745	1.7606	1.7408	1.7029	1.6929	1.6451
	error	0.05%	0.04%	0.03%	0.96%	0.03%	1.33%
6	simulation	1.7754	1.7604	1.7374	1.6994	1.6423	1.5673
	experiment	1.7784	1.7725	1.7328	1.6930	1.6530	1.5774
	error	0.17%	0.69%	0.26%	0.38%	0.65%	0.65%

**Table 10 sensors-19-03232-t010:** Comparison of simulation data and experimentally obtained data of the volumetric corrosion pit.

	Radius (mm)	2	4	6	8	10
Depth (mm)						
1	simulation	1.7664	1.7214	1.6564	1.5813	1.5063
	experiment	1.7407	1.7170	1.6411	1.5796	1.4979
	error	1.46%	0.26%	0.92%	0.11%	0.56%
2	simulation	1.7654	1.7104	1.6243	1.5253	1.4222
	experiment	1.7469	1.6971	1.5934	1.48990.	1.3785
	error	1.05%	0.78%	1.90%	2.32%	3.08%
3	simulation	1.7644	1.7064	1.6113	1.5013	1.3872
	experiment	1.7247	1.6832	1.5915	1.4581	1.3546
	error	2.25%	1.36%	1.23%	2.88%	2.35%
4	simulation	1.7644	1.7034	1.6053	1.4892	1.3682
	experiment	1.7307	1.6851	1.5695	1.4541	1.3544
	error	1.91%	1.07%	2.23%	2.36%	1.01%
5	simulation	1.7644	1.7034	1.6013	1.4832	1.3572
	experiment	1.7327	1.6700	1.5537	1.4462	1.3347
	error	1.80%	1.96%	2.97%	2.50%	1.66%
6	simulation	1.7664	1.7034	1.5983	1.4722	1.3312
	experiment	1.7605	1.6770	1.5854	1.4501	1.2989
	error	0.33%	1.55%	0.81%	1.50%	2.43%

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
