# Peer review of "Influence of Volumetric Damage Parameters on Patch Antenna Sensor-Based Damage Detection of Metallic Structure"

_sensors, 2019, doi:10.3390/s19143232_

Round 1
Reviewer 1 Report
In general, the paper is technically accurate. The objective of the study and the methodology followed are clearly outlined. The conclusions summarize the findings. However, the reviewer has the following questions:
1) Could you describe more in detail the possible practical application of the design sensor in diagnostics of the real structures? What are the limitations of using the sensors described?
2) The damages in the article have regular, circular shape. What in the case of the irregular corrosion pit with varying depth, width and length?
3) The corrosion process usually is associated with a change in material parameters as well as a change in the volume of corrosion products. Could you comment that it is possible to monitor the corrosion damage development in such a case?
Detailed remarks:
1) Tables are difficult to read because of a lot of collected data (for example Table 9 and 10). It is recommended to indicate the smallest and the biggest error to increase the readibility.
2) The article is not written in accordance with the recommendations described in template. Exemplary, in many places the symbols are written using the wrong font and are not written in italics, the main text is wrritten with the use of differerent size font than abstract, etc.
Author Response
Thank you very much for your valuable comments on our manuscript, which really helps to improve our work. We have read the review report carefully, and please see the attachment for the point-to-point responses to issues you listed.
It should be noted that the responses are highlighted in red, the revised content cited from the manuscript is highlighted in blue and the references are highlighted in green. If we did not answer the questions clearly or you have more to discuss, please contact us by this email address: yuhanjinwh@whut.edu.cn.

Reviewer 2 Report
Comments to the Author
This paper presented the simulation analysis and experimental investigation of the volumetric damage identification based on the patch antenna sensor. However, only simulated crack and corrosion pit were investigated and the level of originality is low. The detailed comments regarding the submission are as follows:
1. What is the main contribution of this work to the antenna-sensor-based damage detection? The reviewer finds that the investigations in the submitted paper are previously investigated by many researchers.
2. The abstract and introduction need to be improved.
3. Page 4, line 126, section 3.2.: the authors conducted the FE simulation using COMSOL. However, no convergence study was found. The convergence of the FE model is a critical issue, in terms of the quality of the solution. Also, please add the convergence criteria in your analysis and the detailed information of the FE simulation, such as meshing, element type, and size.
4. For Table 2 and Table 3, the crack length sensitivity is defined as the frequency shift caused by the crack of per unit length. For Table 2, the crack length sensitivity is -22.0110 MHz/mm for the crack depth of 6 mm and crack length of 24 mm, the frequency shift is -528.264 MHz (sensitivity multiply crack length: -22.0110 MHz/mm*24 mm). However, the crack depth sensitivity is -188.0960 MHz/mm for the crack depth of 6 mm and crack length of 24 mm, the frequency shift is -1128.576MHz (sensitivity multiply crack depth: -188.0960 MHz/mm*6 mm). The frequency shift does not match even the crack depth and length are the same. Please explain.
5. Figures 7 and 8 are not clear. Figure 14 (a) and (b) are missing.
6. Table 4 and 5, why the unit of sensitivity is GHz/mm. This does not match your results in Figures 11 and 12.
7. Figure 14 (b), the author pointed out that “the current density along line b abruptly amplifies from zero to the maximum, then gradually decreases to zero.” Please elaborate.
8. The format of Table 9 needs to be adjusted.
9. Table 9 and Table 10: please add an additional row, ‘error’, below ‘experiment’. Please explain how to calculate the error.
Author Response

(The authors gave the same response as above.)

Round 2
Reviewer 1 Report
The paper has been improved. It is suggested to accept this work.
Reviewer 2 Report
The authors improved the manuscript considerably. The authors answered all my questions and introduced appropriate modifications. I do not have any further comments.